# Elevated Antibody Titers to Epstein–Barr Virus and Cytomegalovirus in Patients with Drug-Induced Lupus

**DOI:** 10.3390/v15040986

**Published:** 2023-04-17

**Authors:** Julie Knudsen, Nicole Hartwig Trier, Anette Holck Draborg, Christoffer Tandrup Nielsen, Søren Jacobsen, Peter Højrup, Gunnar Houen

**Affiliations:** 1Department of Biochemistry and Molecular Biology, University of Southern Denmark, 5230 Odense, Denmark; juliekristineknudsen@gmail.com (J.K.); php@bmb.sdu.dk (P.H.); 2Department of Autoimmunity, Statens Serum Institut, 2300 Copenhagen, Denmark; nicole.hartwig.trier@gmail.com (N.H.T.); ahdraborg@gmail.com (A.H.D.); 3Copenhagen Research Center for Autoimmune Connective Tissue Diseases, Center for Rheumatology and Spine Diseases Rigshospitalet, 2100 Copenhagen, Denmark; christoffer.tandrup.nielsen.01@regionh.dk; 4Department of Clinical Medicine, Faculty of Health and Medical Sciences, University of Copenhagen, 2200 Copenhagen, Denmark; soeren.jacobsen.01@regionh.dk

**Keywords:** antibodies, cytomegalovirus, drug-induced lupus, Epstein–Barr virus, systemic lupus erythematosus

## Abstract

Systemic lupus erythematosus (SLE) is an autoimmune disease, which has been associated with Epstein–Barr virus (EBV) and Cytomegalovirus (CMV) infection. Drug-induced lupus (DIL) is a lupus-like disease caused by the intake of therapeutic drugs, which has been estimated to cause approximately 10–15% of lupus-like cases. Although SLE and DIL share common clinical symptoms, there are some fundamental differences between DIL and SLE onset. Moreover, it remains to be examined whether environmental factors, such as EBV and CMV infections, may contribute to the development of DIL. This study focused on examining the possible association between DIL and EBV and CMV infections, by examining IgG titers to EBV and CMV antigens in serum samples by enzyme-linked immunosorbent assays. Antibody titers to EBV early antigen–diffuse and CMV pp52 were found to be significantly elevated in both SLE and DIL patients compared to healthy controls, although no correlation was found for antibodies to the two virus antigens in the respective disease groups. Moreover, total IgG titers were reduced in SLE and DIL serum samples, which may reflect a general lymphocytopenia, which commonly is associated with SLE. The current findings support that EBV and CMV infections may contribute to the development of DIL and that onset of both diseases are related.

## 1. Introduction

Cytomegalovirus (CMV) and Epstein–Barr virus (EBV) are two out of eight human herpesviruses, belonging to the *Herpesviridae* family, which currently comprises about 122 species [1]. The human herpesviruses are highly adapted to their host, and they possess the ability to switch between an active, lytic state and a latent, dormant state, allowing them to survive for long periods of time in the host, as they remain hidden from the immune system [2].

EBV, also known as human herpesvirus (HHV) 4, is a lymphotropic virus and one of the most common human viruses. It was first encountered in cells originating from a patient with African Burkitt’s lymphoma, but later on it was discovered to be found all over the world [3]. EBV infects nearly all individuals (90–95%) at some point in their lives [2,4,5] and has been reported to be associated with numerous diseases [2,6,7,8,9,10]. In the majority of cases, EBV infection takes place in early childhood and typically is asymptomatic. Upon infection, the virus typically remains in latency within the host [2].

Although present in the majority of the population, EBV is only a direct causative agent of various diseases in a few percentages, as EBV infections and reactivations usually are controlled by a well-functioning immune system, which forces EBV back into a dormant state upon infection or reactivation [2].

EBV is estimated to be responsible for approximately 1.5% of all cancers, and EBV has especially been reported to be linked to Hodgkin lymphoma, Burkitt lymphoma and nasopharyngeal cancers [4]. Besides being associated with cancer, EBV is a direct cause of infectious mononucleosis, which primarily affects teenagers and young adults [10]. Moreover, EBV has been proposed to be linked to the onset of a number of autoimmune diseases, such as multiple sclerosis (MS), rheumatoid arthritis (RA), and systemic lupus erythematosus (SLE) [2,7,9,11,12].

Similar to EBV, CMV is a common virus, and the virus is retained in the host for life. Primary CMV infection is often acquired in early childhood and is commonly associated with no symptoms or mild symptoms such as malaise and fever. Over 50% of the population has been infected with CMV by age 40. Similarly to EBV, CMV is associated with infectious mononucleosis; in fact, CMV is the second leading cause of infectious mononucleosis next to EBV [13,14]. In contrast to EBV, which is associated with several severe diseases, most people do not know they are infected with CMV, because the virus seldom causes problems in healthy individuals. Nevertheless, primary CMV infections or reactivations are responsible for significant morbidity and mortality in immunocompromised individuals [15,16,17]. In addition, CMV has been proposed to be involved in the onset of autoimmune diseases, as signs of active viral infection have been found in different autoimmune diseases such as rheumatological and neurological disorders [18].

Similarly, as EBV, CMV infection has been frequently linked to increased production of autoantibodies in various autoimmune diseases, including SLE, where EBV has been reported to be associated with disease onset in genetically predisposed individuals as well [19,20,21,22]. Evidently, SLE patients have dysfunctional control of EBV, resulting in frequent reactivations and disease progression [7]. Collectively, these findings point to the fact that SLE is a multifactorial autoimmune disease, where several factors contribute to disease onset. SLE is a chronic autoimmune disease. Although both CMV and EBV have been reported to act as environmental triggers and contribute to SLE disease onset in genetically predisposed individuals, the etiology of the disease remains unknown [7,19,20,21,22,23]. Patients with SLE experience a variety of symptoms, including fatigue, skin rashes, fever, and pain or swelling of the joints [23]. The disease is characterized by cycles of disease activity, termed flares, and remission, where no symptoms occur. Women are most at risk of developing SLE, as the female-to-male ratio is about 9:1, and disease onset typically peaks between the ages of 18–45 years [23]. The incidence ranges from 40 to 200 per 100.000 individuals per year, depending on geography and ethnic origin [23].

Drug-induced lupus (DIL) is a lupus-like disease, which is caused by the intake of some therapeutic drugs, leading to lupus-like clinical symptoms [24,25,26,27]. Approximately 10–15% of lupus cases are estimated to be caused by the administering of various therapeutic drugs [24]. Hydralazine was the first drug originally described to be associated with the development of lupus-like symptoms [28]. Since then, more than 100 drugs from more than 10 different drug classes have been identified as a possible cause of DIL [24,25,29]. Most known are the high-risk drugs hydralazine and procainamide. Risk rates as high as 30% with procainamide and 5–10% with hydralazine have been reported [24,25]. However, other common drugs such as minocycline, penicillamine, and anti-TNF biologics have also been reported to induce DIL onset [24,30]. As a consequence of being directly related to drug intake, the epidemiology of DIL is directly reflective of the population using the specific drugs; e.g., minocycline, used to treat acne, is reported to induce lupus in younger females, with a mean age of 21, whereas hydralazine- and procainamide-induced lupus is most common in the elderly population [25,31].

Skin rash is one of the most common clinical symptoms in DIL and SLE. In fact, DIL and SLE share many of the same symptoms, e.g., fever, arthralgia, and serositis [23,24]. However, there are also some fundamental differences between the two diseases. For example, the presence of antibodies to dsDNA is very frequent in SLE patients, but it is more rare in DIL patients [32]. Furthermore, the age of DIL onset is often higher compared to SLE, which is associated with the age at which the patient commenced treatment with the particular drug, as previously mentioned. The female-to-male ratio is typically 1:4–1:1, depending on the administered drug, compared to a male-to-female ratio of 9:1 in SLE [23,33]. Moreover, DIL tends to be less severe than SLE, although DIL can be more challenging to diagnose [24,34,35,36]. However, lupus-like symptoms with the exclusion of other autoimmune disorders and the resolution of symptoms with the withdrawal of medications often suggest a diagnosis of DIL, as it usually resolves within a few weeks after discontinuation of the drug [34,35,36]. Consequently, DIL carries a favorable prognosis with less morbidity and mortality when compared to SLE.

In addition to environmental triggers, genetics have been proposed to influence DIL onset as well [36,37,38,39]. Nevertheless, no specific pattern has been established, as dependency of genetic risk factors such as HLA-DR4, HLA-DR0301, and complement C4 null allele appear to vary between different agents [36,37,38,39]. Hence, a multifactorial etiology may apply to both DIL and SLE, as only a small fraction of patients receiving drugs develops DIL.

In the current study, we analyzed the presence of selected virus antibodies in DIL and SLE patients, in order to determine whether specific virus infections are associated with onset of DIL.

## 2. Materials and Methods

### 2.1. Materials

Recombinant early antigen–diffuse (EA/D), Epstein–Barr virus nuclear antigen (EBNA)1 (mosaic, residues 1-90/408-498), and CMV pp52 were purchased from Protein Specialists (Ness-Ziona, Israel). Purified human intravenous IgG (IVIG) [40], Tris-Tween-NaCl (TTN) buffer (0.25 M Tris, 0.5% Tween 20, 0.15 M NaCl, pH 7.5), alkaline phosphatase (AP)-substrate buffer (1 M diethanolamine, 0.5 mM MgCl_2_, pH 9.8), carbonate buffer (50 mM sodium carbonate, pH 9.6) were from Statens Serum Institut (Copenhagen, Denmark). AP-conjugated goat anti-human IgG and AP-substrate tablets (*p*-nitrophenyl phosphate (*p*NPP)) were purchased from Sigma Aldrich (St. Louis, MO, USA). Polysorp microtiter plates were from NUNC/Thermo Fisher (Roskilde, Denmark).

### 2.2. Patient Sera

Sera from patients with DIL or drug-induced vasculitis (DIV), as clinically diagnosed by the treating physicians, were obtained from the Biobank at Statens Serum Institut (Copenhagen, Denmark). DIV samples were used as negative lupus controls. DIL and DIV sera were used anonymously, and hence no personal information about these sera were available. However, the samples are expected to be a representative of the Danish population and to follow already reported common characteristics for individuals with DIL and DIV in relation to age, gender, and serologic characteristics [33]. Sera from SLE patients and age- and sex-matched healthy controls (HCs) were collected at the Copenhagen Research Center for Autoimmune Connective Tissue Diseases, Center for Rheumatology and Spine Diseases, Rigshospitalet (Copenhagen, Denmark). SLE patients were diagnosed according to the revised American College of Rheumatology 1997 classification criteria [41]. All samples enrolled were obtained from Caucasian contributors. All SLE sera studied were positive for antibodies to dsDNA, whereas 7.5% of the DIL samples were positive for autoantibodies to dsDNA (Table 1). A pool of healthy blood donors (HD) was made of 100 anonymous samples obtained from the Blood Bank at Rigshospitalet (Copenhagen, Denmark), functioning as a control. All samples were used in accordance with the relevant ethical guidelines, and the use of clinical samples was approved by the Ethical Committee of Copenhagen (No. H-A-2007-0114).

### 2.3. Quantification of Antibodies in Serum by Enzyme-Linked Immunosorbent Assay

A direct enzyme-linked immunosorbent assay (ELISA) was conducted to determine CMV pp52 and EBV EA/D IgG levels in patient sera and control sera. Coating of the Polysorp microtiter plates was performed by using diluted recombinant EBV EA/D, EBV EBNA1, and CMV pp52 proteins (1 µg/mL) in carbonate buffer overnight at 5 °C (100 µL/well), followed by washing (3 × 5 min) and blocking for 1 h at room temperature (RT) with TTN buffer. The TTN buffer was used for all dilutions and incubation steps with sera and secondary antibodies (conjugates), for washing and for blocking steps. Diluted patient samples and HC samples (1:100) were added to coated and non-coated wells in duplicate, and then incubated for 1 h at RT. Following rinsing of the plates, AP-conjugated goat anti-human IgG (1:2000) was added to the wells, whereafter the plates were incubated for 1 h on a shaking table. Next, the plates were washed, whereafter AP-substrate buffer (1 mg/mL) was added. After an appropriate color development (approximately 30 min), the absorbance was measured at a wavelength of 405 nm with a reference wavelength of 650 nm on a Versamax microplate reader (Molecular Devices, Sunnyvale, CA, USA).

On each plate, twofold dilution curves were made using a HD pool in order to generate a standard curve for normalizing the results of the patient sera (U/mL). The standard curve was produced on the basis of absorbances corresponding to the following dilutions: 1:10, 1:20, 1:40, and 1:80, and 1:50, 1:100, 1:200, and 1:400. A SLE pool (*n* = 30) was used as a positive control, whereas a HC pool (*n* = 40) was used as a negative control. Blank wells were used for background determination and were subtracted prior to data analysis. All samples were tested in duplicate.

### 2.4. Quantification of Total IgG in Serum Samples

A competitive ELISA was performed on Maxisorp plates coated with IVIG (1 µg/mL PBS, 100 µL/well) ON at 5 °C. The wells were washed 3 × 1 min with TTN buffer. AP-conjugated goat anti-human IgG (1:1000) was used for the generation of two standard curves; one with the IgG standard diluted in PBS to 20, 16, 14, 12, 10, 8, and 6 mg/mL, respectively. All were diluted (1:100) in PBS (1 µg/mL). The other standard curve was produced as a 10-fold dilution, which was made from a 10 mg/mL stock solution of IgG standard (IVIG, 50 mg/mL) [40]. All samples were preincubated for 1 h with goat anti-human IgG (1:1000) before transfer to the IgG-coated plate. A HD pool was used for both high and low positive controls; the low control was diluted 1:1 in human albumin (50 mg/mL), and the high control was used non-diluted. Individual serum samples (1:100) were incubated with goat anti-human IgG (1:1000). All samples were tested in duplicate and were preincubated for 1 h at RT in the incubation solution together with IgG conjugate (1:1000), while the wells were blocked with TTN buffer. After incubation and another 3 washes with TTN, AP-substrate buffer was added, and the plates were read after 1 h at 405 nm and a reference wavelength of 650 nm, as described above.

### 2.5. Statistics

Calculations were carried out using the statistical software R Studio (RStudio, Boston, MA, USA). ANOVA was used to compare the groups. A *p*-value lower than or equal to 0.05 was considered significant. Significant differences are indicated by *: *p* < 0.05, **: *p* < 0.01, ***: *p* < 0.001. For correlation analysis of the results, the Pearson’s r and *p* values were determined for each analysis. Correlations were classified accordingly: r: 0.0–0.19 very low, 0.2–0.39 low, 0.4–0.59 moderate, 0.6–0.79 high, 0.8–1.0 strong correlation.

## 3. Results

### 3.1. Detection of EA/D IgG and CMV pp52 IgG in Serum Samples of Lupus-Associated Diseases and Healthy Controls

EBV EA/D and CMV pp52 IgG levels were determined in individual serum samples by ELISA. In total, 40 HCs, 30 SLE, and 40 DIL samples were tested for reactivity to CMV pp52 (Figure 1a). Moreover, the same samples and also 20 DIV samples were tested for reactivity to EBV EA/D as well (Figure 1b).

Figure 1a illustrates the reactivity of SLE and DIL patient sera and HC sera to CMV pp52. CMV pp52 IgG levels were significantly elevated in DIL and SLE sera when compared to the HCs (*p* = 0.05 and 0.01, respectively). Similarly, EBV-EA/D IgG in SLE and DIL sera was elevated compared to HC sera as well (*p* = 0.004 for SLE, and *p* = 0.03 for DIL) and DIV sera (*p* < 0.0001) (Figure 1b).

Based on elevated IgG titers to both EBV EA/D and CMV pp52 in DIL and SLE serum samples compared to HCs, an IgG correlation analysis between the virus-specific IgG levels was conducted in the two disease groups (Figure 2). However, as seen, no correlation was determined between EBV EA/D IgG and CMV pp52 IgG in SLE and DIL samples, as r scores of 0.08943 and −0.01628 were obtained, respectively.

### 3.2. Corrected IgG Concentrations

Next, the total IgG levels were determined in individual DIL, DIV, SLE, and HC serum samples by competitive ELISA (Figure 3). This was done to verify if the statistically significant difference between patient samples and HCs was caused by a higher IgG level or whether the effect was general.

As seen, the total IgG levels were significantly reduced in DIL samples compared to HCs (*p* = 0.0048) and SLE samples (*p* = 0.0177) (Figure 3). Moreover, a trend indicated that total IgG levels in SLE samples were also slightly reduced compared to HCs, although this was not significant (*p* = 0.1409). Finally, the total IgG concentrations in DIV samples were significantly reduced relative to HCs (*p* = 0.0130), but not when compared to DIL and SLE samples (*p* > 0.05), although a trend indicated reduced total IgG levels in DIV samples compared to SLE samples.

Following this, antibody concentrations normalized relative to mg IgG were determined. As presented in Figure 4, the pattern already presented (Figure 2) appeared to be independent of the total IgG concentrations, as statistically significantly elevated EBV EA/D IgG and CMV pp52 IgG were detected in serum samples from DIL and SLE patients compared to HCs (EBV EA/D IgG, *p* = 0.002 for SLE vs. HC, *p* = 0.04 for DIL vs. HC) (CMV IgG *p* = 0.03 for SLE vs. HC, *p* = 0.01 for DIL vs. HCs). Similarly, significantly reduced EBV EA/D IgG concentrations were identified for DIV samples when compared to SLE samples (*p* = 0.001).

## 4. Discussion

In the present study, we determined IgG reactivities to antigens from EBV (EBNA1) and CMV (pp52) in DIL, SLE, and HC serum samples. Initial screenings showed significantly elevated IgG levels to EBV EA/D and CMV pp52 in DIL samples compared to HCs (Figure 1). Similarly, CMV pp 52 IgG and EBV EA/D IgG were significantly elevated in SLE samples compared to HCs. In contrast, no notable differences in EBV EA/D IgG levels in the DIV samples were determined when compared to the HCs. Although CMV pp52 and EBV EA/D IgG levels were elevated in both SLE and DIL compared to HCs, no significant differences in these IgG levels were determined between the two lupus groups, and no correlation was determined between the CMV pp52 IgG and EBV EA/D IgG in SLE and DIL groups (Figure 2).

As presented, total IgG levels in the respective samples were determined (Figure 3), indicating that patients with DIL and SLE experience slightly reduced IgG levels compared to HCs, which may simply reflect the general lymphopenia seen in SLE and is in accordance with a current theory of EBV being involved in DIL and SLE onset, as EBV-infected B cells are prone to elimination by cytotoxic T cells and NK cells [7,42].

Based on total IgG concentrations, corrected (normalized) EBV EA/D IgG and CMV pp52 IgG levels were determined, which confirmed the initial findings that EBV EA/D IgG and CMV pp52 IgG levels were elevated relative to the HC group (Figure 4). DIV samples functioned as a negative control relative to DIL samples and as presented; no significant difference in EBV EA/D IgG was found in DIV samples relative to DIL nor HC samples (Figure 1 and Figure 4).

Based on these observations, the current findings may indicate that elevated EBV EA/D and CMV pp52 IgG levels are not simply a result of general B-cell hyperactivity.

To our knowledge, this is the first study demonstrating an association between viral EBV and CMV infections and DIL. A potential caveat in the study is the uncertainty of the DIL diagnosis. When obtaining sera from DIL patients, they obviously may have another underlying disease, which is the cause of drug treatment. Another weakness is that we do not know which drug caused DIL onset in individual patients, since there may be some differences in the action of the various drugs. Similarly, it has not been possible to match SLE and DIL samples according to gender and age; however, the samples are expected to be a representative selection of the Danish population and to follow already reported common characteristics for individuals with DIL and DIV in relation to age, gender, and serologic characteristics, as already mentioned [33].

In a study by Hanlon et al., observations from 568 SLE patients and 368 controls were compared, and the authors reported a highly significant difference in IgG antibodies against EBV EA/D between SLE patients and HC, which supports the results on SLE patients found in this study [20]. Larsen et al. found no significantly altered CMV-specific responses in SLE patients [43], while Dubey et al. [21] and Rasmussen et al. [19] reported a significantly higher seroprevalence of antibodies to certain CMV antigens in SLE patients compared to controls, but insufficient to establish a definite role in SLE onset.

The possible reasons for the ambiguity in different studies could be that the results may be influenced by the type of assay used for measuring antibodies to the various antigens and also by which patients are tested. CMV pp52 is a protein that is vital for lytic replication of CMV [44], which also applies to EA/D in relation to the EBV lytic cycle [21,45]. The aim in this study was to measure antibodies to antigens reflecting the lytic virus stage and to compare with previous results obtained for SLE patients. We found a significant association between high levels of such antibodies and a diagnosis of DIL. In theory, lymphopenia would be expected to result in lower levels of antibodies, including specific EBV and CMV antibodies and also in lower levels of T cells in general. It has previously been reported that SLE patients frequently suffer from immune deficiencies such as general lymphopenia and/or fewer (specific) cytotoxic CD8^+^ T cells. In four previously published studies, the authors reported a decrease in T-cell response when blood samples were stimulated with EBV antigens, and fever cytotoxic CD8^+^ T cells in general in SLE patients [20,42,46,47]. It is yet to be examined, if these deficiencies also apply to DIL patients; however, it seems likely that some genetic immune deficiency precedes the disease, as not all individuals receiving treatment with DIL-inducing drugs actually develop DIL. An unresolved question is whether a preceding infection with the virus precipitates disease development, or if DIL patients harbor a poorly functioning immune system, leading to reactivation of EBV and CMV.

Larsen et al. suggested that reactivation of EBV seems to be a consequence of a preexisting immune deficiency, rather than the cause of SLE immunopathology, a consequence that has an exacerbating effect on the process of immune activation in SLE patients [44]. It seems a fair notion that immune deficiency precedes EBV reactivation. However, it is also possible that the cause of SLE, as facilitated by immune defects and EBV infection, is in fact autoantibodies produced as a response to the apoptotic waste load caused by increased apoptosis of infected lymphocytes by reactivated EBV [48,49]. This could be working in conjunction with other EBV mechanisms of action, such as expression of immune-modulating proteins [48,50]. A hitch in the conclusion by Larsen et al. [43] is also that you cannot have one without the other. Lacking the immune deficiency, the EBV would not be able to reactivate in an uncontrollable manner, leading to the production of auto-antibodies and thereby autoimmunity, but without EBV, the immune deficiency does not induce autoimmunity, which makes EBV a vital agent in the development of certain autoimmune diseases.

No matter how EBV induces autoimmunity in SLE patients, it seems that the course of action is slightly different in DIL patients. The most obvious explanation for DIL pathogenesis is that individuals with a poor immune response to EBV infections, but not to a degree as to develop SLE, may develop DIL because of the action of certain therapeutic drugs, which may be able to reactivate dormant EBV and/or CMV.

In conclusion, it was found that there is a strong implication for a significant relationship between EBV, CMV, and DIL, as evidenced by increased levels of antibodies to virus lytic cycle antigens. This study is small, but the results provide a basis for a larger prospective study.

## Figures and Tables

**Figure 1 viruses-15-00986-f001:**
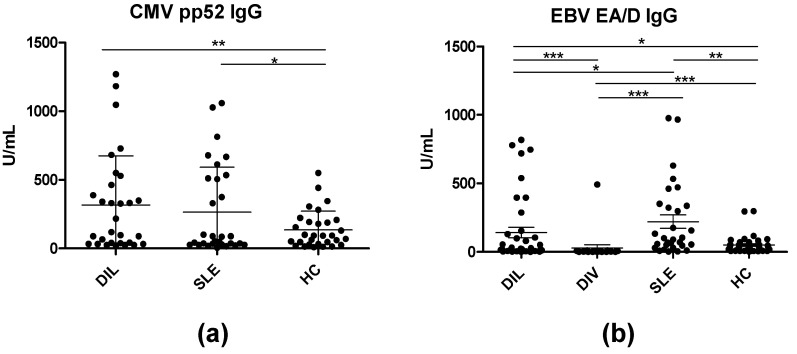
CMV pp52 and EBV EA/D IgG levels in patient samples and healthy controls analyzed by ELISA. (**a**) Reactivity of DIL, SLE, and HC sera to CMV pp52. (**b**) Reactivity of DIL, DIV, SLE, and HC sera to EBV EA/D. Significant differences are indicated by *: *p* < 0.05, **: *p* < 0.01, ***: *p* < 0.001.

**Figure 2 viruses-15-00986-f002:**
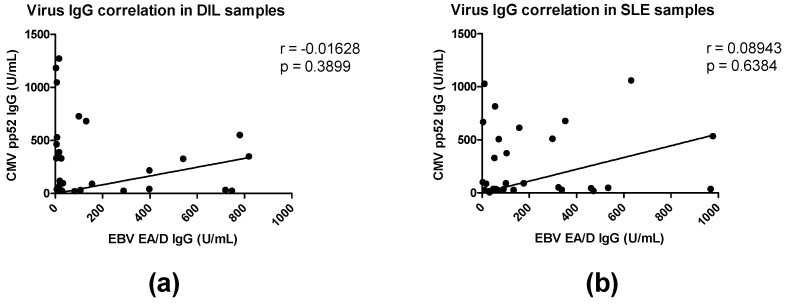
Correlations of EBV and CMV IgG levels in SLE and DIL patient sera. (**a**) Correlation between CMV pp52 IgG and EA/D IgG in DIL patient sera. (**b**) Correlation between CMV pp52 IgG and EA/D IgG in SLE patient sera.

**Figure 3 viruses-15-00986-f003:**
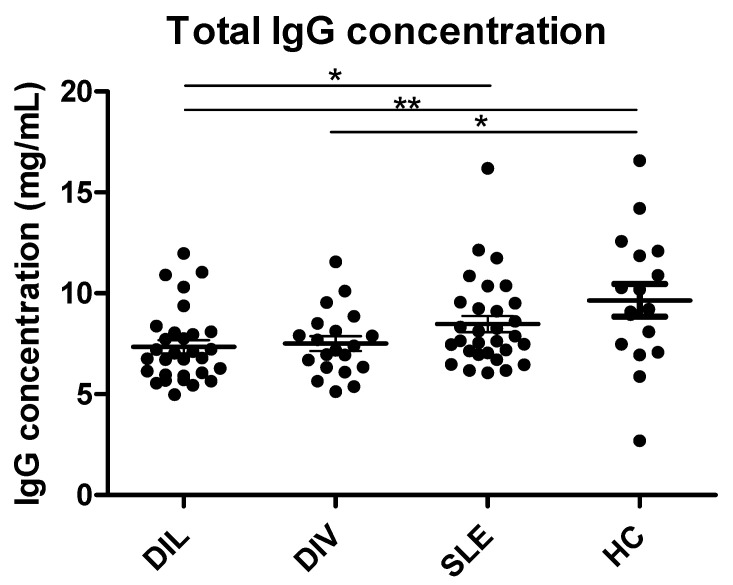
Total IgG concentrations analyzed in DIL, DIV, SLE, and HC sera by inhibition ELISA. Significant differences are indicated by *: *p* < 0.05, **: *p* < 0.01.

**Figure 4 viruses-15-00986-f004:**
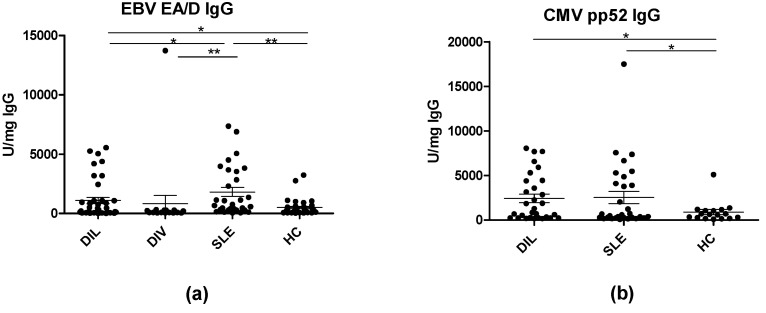
Corrected EBV EA/D and CMV pp52 IgG in serum samples. (**a**) Corrected EBV EA/D IgG in DIL, DIV, SLE, and HC samples. (**b**) Corrected CMV pp52 IgG in DIL, SLE, and HC samples. Significant differences are indicated by *: *p* < 0.05, **: *p* < 0.01.

**Table 1 viruses-15-00986-t001:** Characteristics of applied patient and healthy control sera.

	SLE	DIL	DIV	HCs
No. of individuals	30	40	20	40
Average age (years) [range]	38.4 [22–65]	-	-	38.5 [25–72]
% females	96	-	-	78
% ds DNA antibody-positive	100	7.5	0	0

## Data Availability

All data available are presented in this manuscript.

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
