# Peer review of "Elevated Antibody Titers to Epstein–Barr Virus and Cytomegalovirus in Patients with Drug-Induced Lupus"

_viruses, 2023, doi:10.3390/v15040986_

Round 1

Reviewer 1 Report

Dear Authors,

Please improve and extent your introduction

Please add more conclusion to your article

Kind regards

Author Response

Dear reviewer,

thank you for your comments. We agree that the introduction and conclusion are short. Both sections have been elaborated.

Reviewer 2 Report

This is an interesting study that attempts to explore the association of EBV and CMV to SLE and DIL. The researchers made use of ELISA to look into IgG titers for EBV and CMV antigens. My only concern though is that it would have been better if the researchers further characterized the specific IgG isotype that was elevated in response to EBV early antigen diffuse and CMV pp52 . I also suggest that for the introduction, the researchers should also include a short paragraph discussing the role of EBV and CMV in various diseases like cancer and other autoimmune diseases  before diving into SLE and DIL. DOI: 10.1007/978-3-319-22822-8_15; DOI: 10.1097/BOR.0000000000000289; DOI: 1010.3390/cancers13153909

Author Response

Dear reviewer,

thank you for your constructive comments. We acknowledge that the IgG isotype reacting to EBV and CMV antigens remains to be determined, we are currently planning an expanded study, where IgG isotypes will be determined as well.

The introduction has been elaborated to include the role of EBV and CMV in other diseases as well. Thank you for the references, which has been added to the introduction. We have extended the discussion as well.